# Conservation of transcription factor binding specificities across 600 million years of bilateria evolution

Kazuhiro R Nitta[1], Arttu Jolma[1,2], Yimeng Yin[1], Ekaterina Morgunova[1], Teemu Kivioja[2], Junaid Akhtar[3], Korneel Hens[4], Jarkko Toivonen[5], Bart Deplancke[6], Eileen E M Furlong[3], Jussi Taipale[1,2]*

[1]Department of Biosciences and Nutrition, Karolinska Institutet, Stockholm, Sweden; [2]Genome-Scale Biology Program, University of Helsinki, Helsinki, Finland; [3]Genome Biology Unit, European Molecular Biology Laboratory, Heidelberg, Germany; [4]Institute of Bioengineering, School of Life Sciences, Swiss Federal Institute of Technology, Lausanne, Switzerland; [5]Department of Computer Science, University of Helsinki, Helsinki, Finland; [6]Institute of Bioengineering, School of Life Sciences, Ecole Polytechnique Fédérale de Lausanne, Lausanne, Switzerland

**Abstract** Divergent morphology of species has largely been ascribed to genetic differences in the tissue-specific expression of proteins, which could be achieved by divergence in *cis*-regulatory elements or by altering the binding specificity of transcription factors (TFs). The relative importance of the latter has been difficult to assess, as previous systematic analyses of TF binding specificity have been performed using different methods in different species. To address this, we determined the binding specificities of 242 *Drosophila* TFs, and compared them to human and mouse data. This analysis revealed that TF binding specificities are highly conserved between *Drosophila* and mammals, and that for orthologous TFs, the similarity extends even to the level of very subtle dinucleotide binding preferences. The few human TFs with divergent specificities function in cell types not found in fruit flies, suggesting that evolution of TF specificities contributes to emergence of novel types of differentiated cells.

*For correspondence: jussi.taipale@ki.se

**Competing interests:** The authors declare that no competing interests exist.

**Reviewing editor**: Bing Ren, University of California, San Diego School of Medicine, United States

## Introduction

It is estimated that the divergence between vertebrate and invertebrate lineages occurred over 600 million years ago (*Hedges et al., 2006*; *Peterson et al., 2008*). After the divergence, protein coding sequences have retained a relatively high level of similarity, whereas homology in gene-regulatory elements is not detectable, despite the fact that many developmental pathways and regulatory relationships between TFs and their target genes have been conserved (see e g., *Goodrich et al., 1996*; *Pichaud and Desplan, 2002*; *Ciglar and Furlong, 2009*). Lack of sequence conservation in gene regulatory elements despite their conserved function could be a consequence of divergence of the gene regulatory code between vertebrates and invertebrates. Some changes in coding sequences of TFs have indeed been observed, and linked to specific evolutionary adaptations (*Enard et al., 2002*; *Di-Poï et al., 2010*).

Several studies have indicated that primary TF DNA binding specificity evolves slowly, and is extremely conserved between mammalian species (see e g., *Bohmann et al., 1987*; *Struhl, 1987*; *Merika and Orkin, 1993*; *Amoutzias et al., 2007*; *Wei et al., 2010*; *Jolma et al., 2013*). The origin of most structural families of TFs dates well before the emergence of mammals, and even predates the

**eLife digest** Flies look very different from humans, but both are descended from a common ancestor that existed over 600 million years ago. Some differences between animal species are due to them having different genes: stretches of DNA that contain the instructions to make proteins and other molecules. However, often differences are caused by the same or similar genes being switched on and off at different times and in different tissues in each species.

The instructions that control when and where a gene is expressed are written in the sequence of DNA bases located in the regulatory region of the gene. These instructions are written in a language that is often called the 'gene regulatory code'. This code is read and interpreted by proteins called transcription factors that bind to specific sequences of DNA (or 'DNA words') and increase or decrease gene expression. Changes in gene expression between species could therefore be due to changes in the transcription factors and/or changes in the instructions within the regulatory regions of specific genes.

Gene regulatory regions are not well conserved between species. However, it is unclear if the instructions in these regions are written using the same gene regulatory code, and whether transcription factors found in different species recognize different DNA words.

Nitta et al. have now used high-throughput methods to identify the DNA words recognized by 242 transcription factors from a fruit fly called *Drosophila melanogaster*. Nitta et al. then used new computational tools to find motifs, or collections of DNA words, that are recognized by each of the transcription factors. By comparing the motifs, they observed that, in spite of more than 600 million years of evolution, almost all known motifs found in humans and mice were recognized by fruit fly transcription factors.

Nitta et al. noted that both fruit flies and humans have transcription factors that recognize a few unique motifs, and confer properties that are specific to each species. For example, some of the transcription factors that control the development of the fruit fly wing are not present in humans. Moreover, fruit flies lack both mucus-producing goblet cells and the ability to recognize a motif read by the transcription factor that controls the development of these cells in humans.

The findings of Nitta et al. also indicate that transcription factors do not evolve to recognize subtly different DNA motifs, but instead appear constrained to recognize the same motifs. Thus, much like the genetic code that instructs how to build proteins, the gene regulatory code that determines how DNA sequences direct gene expression is also highly conserved in animals. The language used to guide the development of animals has, as such, remained very similar for millions of years. What makes animals different is differences in the content and length of the instructions that are written using this language into the regulatory regions of their genes.

divergence of vertebrates and invertebrates. Within each TF family, DNA binding specificity has also diverged considerably, with many families having 2–10 different subclasses displaying different primary binding specificities (*Berger et al., 2008*; *Wei et al., 2010*; *Jolma et al., 2013*). In addition, many differences in the TF repertoire between invertebrates and vertebrates exist due to the expansion of some gene families, such as nuclear receptors and C2H2 zinc finger factors in vertebrates (*Charoensawan et al., 2010*). It is currently unclear to what extent such expansion of TF families has changed the gene regulatory code by introducing novel DNA-binding specificities.

Systematic comparison of DNA binding specificities between vertebrates and invertebrates has been difficult. Databases collecting TF binding specificity information, such as TRANSFAC (*Matys et al., 2006*) and Jaspar (*Bryne et al., 2008*) contain a large number of specificities from different species. However, the data are generally derived using different methods in different laboratories, and therefore it is very difficult to separate method-specific biases from real differences in binding specificity, particularly in cases where the differences are not very pronounced. Comparison of TF binding specificities obtained using in vivo methods such as ChIP-seq or deep DNase I hypersensitivity (*Celniker et al., 2009*; *ENCODE Project Consortium, 2012*; *Wang et al., 2013*; *He et al., 2014*), in turn, can result in identification of signals derived from heterodimers or indirect DNA binding, and is affected by biases due to the fact that different genomes have different GC content and repertoire of repetitive elements. Furthermore, although large systematic datasets exist for in vitro binding

specificity of human and mouse TFs determined using protein-binding microarrays (*Badis et al., 2009*) and high throughput SELEX (HT-SELEX; *Jolma et al., 2013*), these methods have not been previously used to systematically analyze specificities of many TF families from more distantly related organisms. Conversely, a large collection of *Drosophila* TF binding specificities has been generated using bacterial one-hybrid assay (B1H; *Noyes et al., 2008*; *Zhu et al., 2011*; *Enuameh et al., 2013*). However, this method has not been applied to mammalian TFs at a scale sufficient for a broad comparison of TF specificity.

The lack of large, high resolution datasets generated using similar methods have limited previous studies of conservation of TF binding specificity to studies of divergence of strong core binding specificities in a limited number of cases (see e g., *Bohmann et al., 1987*; *Struhl, 1987*). However, in addition to the primary specificity, many TFs have secondary modes of DNA binding, can recognize multiple different homodimeric sites and/or recognize multiple different sequences based on DNA shape (*Badis et al., 2009*; *Rohs et al., 2010*; *Jolma et al., 2013*). These, more subtle DNA binding preferences could potentially change faster than and independently of the canonical core DNA recognition motif, and could thus also allow TF specificity to slowly drift in evolution.

To characterize the role of divergence and subtle drift in TF DNA binding specificity in animal diversity, we used HT-SELEX to determine TF binding specificities in *Drosophila melanogaster*. For this, we used existing full-length collection of *Drosophila* TF cDNAs (*Hens et al., 2011*), and also generated a novel genome-scale collection of fruit fly TF DNA binding domain (DBD) constructs. The TFs were expressed in *E. coli* and their specificity determined using HT-SELEX (*Jolma et al., 2010*, *2013*), and compared to our existing human and mouse HT-SELEX data. A striking level of conservation of TF binding specificity was observed, with fruit flies having almost as complex a repertoire of TF binding motifs as humans. Conservation of specificity extended to secondary modes of binding, and even subtle dinucleotide preferences, suggesting that TF binding specificities are not subject to substantial evolutionary drift. Specificities exclusive to either of the species were detected most commonly in cases where a TF subfamily had expanded in one of the species. Only two clear cases of divergence of specificity were observed between orthologs. These results indicate that TF evolution is constrained by structural limitations of the TF folds, and that changes in specificity are rare, and when they occur, tend to have relatively large effects. Interestingly, TF specificities that exist only in human are important for physiology of cell types that do not exist in fruit flies, suggesting that novel TF specificities contribute to formation of new types of differentiated cells.

## Results

### Determination of DNA binding specificities of *Drosophila* TFs by HT-SELEX

A collection of cDNAs of DNA binding domains (DBDs) and full-length *Drosophila* TFs were compiled based on annotations in FlyTF.org (*Pfreundt et al., 2010*) and *Hens et al. (2011)*, respectively. A total of 760 DBDs and 633 full length fruit fly TF constructs were expressed in *E. coli* as N-terminal thioredoxin-HIS tag fusion proteins, and subjected to the HT-SELEX process (*Figure 1*; *Supplementary file 1A–C*; see 'Materials and methods' for details). Together, the clone collections covered 271/294 (92%) and 612/754 (81%) of 'trusted TFs' and 'putative TFs' in FlyTF.org (*Pfreundt et al., 2010*). Briefly, the expressed proteins were purified, incubated with a selection ligand library containing 20 or 40 base pair random DNA sequences, and immobilized to nickel agarose beads. Co-bound ligands were then separated from free ligands, PCR amplified, and the process repeated 3–6 times. Initial selection ligand libraries, and libraries obtained after each selection cycle were sequenced using a massively parallel sequencer (HT-SELEX; *Jolma et al., 2010*, *2013*).

The obtained sequences were analyzed computationally, resulting in the identification of robustly enriched subsequences for a total of 196 DBDs and 92 full-length TF proteins, representing a total of 242 unique fruit fly TFs. We next determined the primary binding profiles for each protein by aligning the enriched sequences to a seed sequence, and calculating a position weight matrix (PWM) using the background corrected multinomial method as described previously (*Jolma et al., 2010*, *2013*). Motifs were obtained for 86% of fruit fly TF families, with median within-family coverage of 49% (*Figure 1B*). Similar to earlier studies (*Badis et al., 2009*; *Jolma et al., 2013*), a relatively low success rate was observed for C2H2 zinc fingers and basic helix-loop-helix proteins, probably at least in part due to difficulty in computational identification of DNA-binding zinc finger proteins, and the fact that many

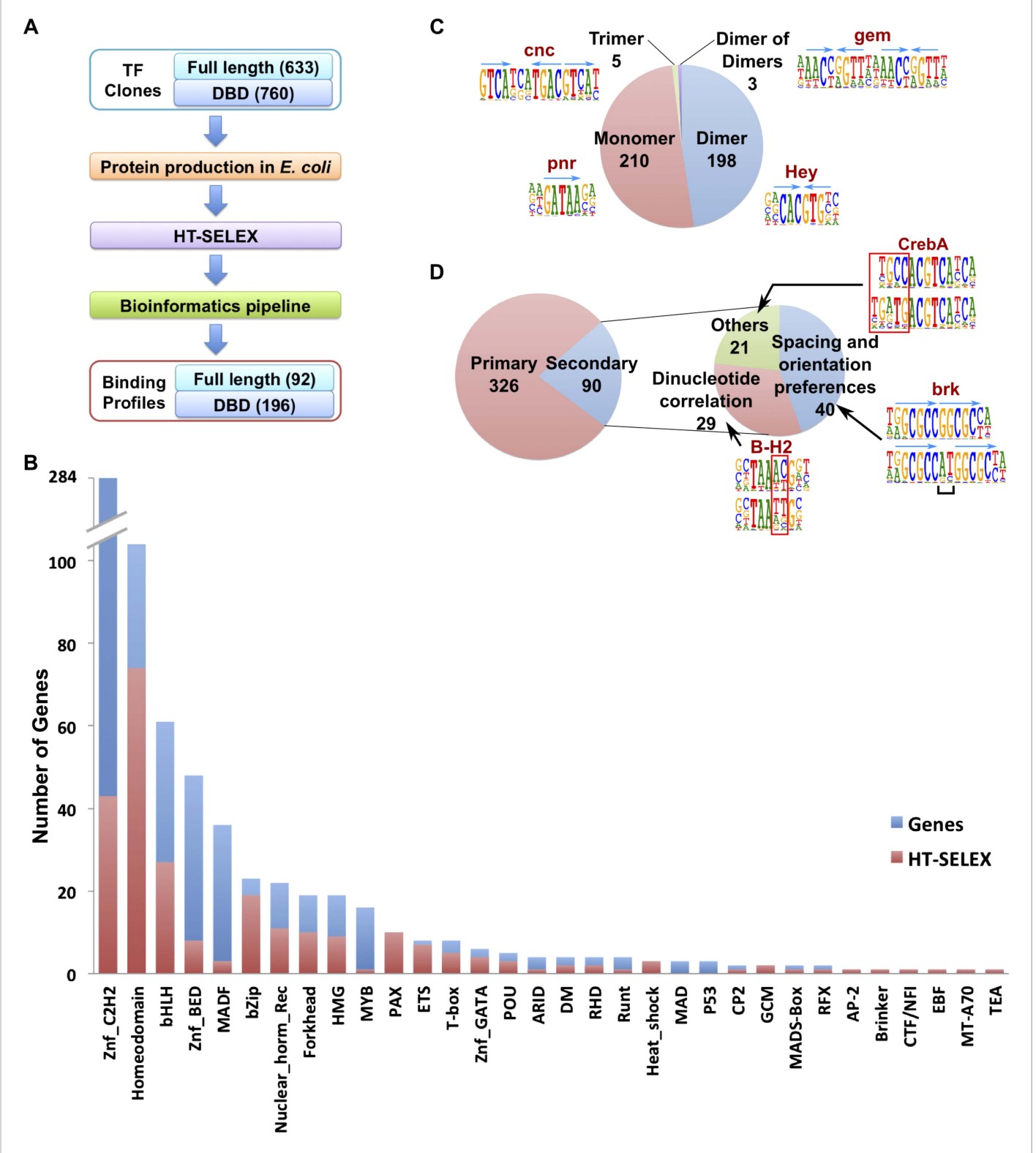

**Figure 1**. *Drosophila* HT-SELEX. (**A**) *Drosophila* HT-SELEX pipeline. (**B**) Coverage of TFs. Number of genes (blue), and number of genes for which we obtained a HT-SELEX model (red) are shown for each TF structural family. Genes that encode more than one domain are counted as members of multiple structural categories. (**C**) Classification of all binding models into non-repetitive sites (monomer), and sites with two, three or four similar subsequences (dimer, trimer and dimer of dimers, respectively). Logos, for an example, for each type of model are shown, arrows indicate half-sites to highlight multimeric sites. (**D**) Classification of primary and secondary models (left). Type of secondary model is indicated on the right. Red boxes and black bracket

*Figure 1. continued on next page*

*Figure 1. Continued*

indicate differences between the primary (top) and secondary (bottom) models. Seeds for the generation of the models were identified using the Autoseed algorithm (see 'Material and methods' and *Figure 1—figure supplements 1–14* for details).

The following figure supplements are available for figure 1:

**Figure supplement 1**. Examples of subsequences that have a Huddinge distance of one.

**Figure supplement 2**. Amino-acid sequence similarity dendrograms for major TF families (human, mouse and *Drosophila*) annotated with bHLH motifs obtained using HT-SELEX.

**Figure supplement 3**. Amino-acid sequence similarity dendrograms for major TF families (human, mouse and *Drosophila*) annotated with bZIP motifs obtained using HT-SELEX.

**Figure supplement 4**. Amino-acid sequence similarity dendrograms for major TF families (human, mouse and *Drosophila*) annotated with Ets motifs obtained using HT-SELEX.

**Figure supplement 5**. Amino-acid sequence similarity dendrograms for major TF families (human, mouse and *Drosophila*) annotated with Fox motifs obtained using HT-SELEX.

**Figure supplement 6**. Amino-acid sequence similarity dendrograms for major TF families (human, mouse and *Drosophila*) annotated with HMG motifs obtained using HT-SELEX.

**Figure supplement 7**. Amino-acid sequence similarity dendrograms for major TF families (human, mouse and *Drosophila*) annotated with Homeobox motifs obtained using HT-SELEX.

**Figure supplement 8**. Amino-acid sequence similarity dendrograms for major TF families (human, mouse and *Drosophila*) annotated with IPT/TIG motifs obtained using HT-SELEX.

**Figure supplement 9**. Amino-acid sequence similarity dendrograms for major TF families (human, mouse and *Drosophila*) annotated with Nuclear receptor motifs obtained using HT-SELEX.

**Figure supplement 10**. Amino-acid sequence similarity dendrograms for major TF families (human, mouse and *Drosophila*) annotated with Pax (based on paired box) motifs obtained using HT-SELEX.

**Figure supplement 11**. Amino-acid sequence similarity dendrograms for major TF families (human, mouse and *Drosophila*) annotated with Tbox motifs obtained using HT-SELEX.

**Figure supplement 12**. Amino-acid sequence similarity dendrograms for major TF families (human, mouse and *Drosophila*) annotated with Zf-GATA motifs obtained using HT-SELEX.

**Figure supplement 13**. Amino-acid sequence similarity dendrograms for major TF families (human, mouse and *Drosophila*) annotated with Zf-C2H2 motifs obtained using HT-SELEX.

**Figure supplement 14**. Reproducibility of the HT-SELEX pipeline.

bHLH proteins bind to DNA as obligate heterodimers. In addition, relatively low success rate was observed for large DBD constructs and full-length proteins, probably due to difficulty of expressing them in an active form in *E. coli*.

## Identification of secondary binding modes and dinucleotide preferences

To improve the detection of enriched subsequences in the HT-SELEX experiments, we developed a novel algorithm, Autoseed, that identifies all subsequences that represent local maxima, that is, are enriched more than any closely related sequence. The method is based on a novel distance measure between two subsequences, the Huddinge distance (see *Figure 1—figure supplement 1* and

'Material and methods' for details). In this method, a subsequence is aligned to all other subsequences, and its count is compared to counts of subsequences that contain $n - 1$ perfectly matching bases, where $n$ is the maximum number of defined bases in the aligned subsequences. The method compares also gapped subsequences to ungapped ones, and can thus identify diverse motifs with widely spaced recognition sites, and differentiate between monomers and dimers of the same subsequence. In addition, as subsequences that differ by more than one substitution are not compared to each other, the method can identify strong dinucleotide preferences. Subsequences that differ by one in the number of defined bases are also compared to each other using a threshold to correct for the higher expected count of the subsequence that contains lower number of defined bases.

Analysis of the fruit fly HT-SELEX dataset using Autoseed resulted in the identification of 416 motifs, of which 210 were monomeric, 198 dimeric, 5 trimeric, and 3 dimers of dimers (*Figure 1C*). Of these motifs, 90 describe secondary binding modes—bound sequences that are distinctly different from the primary TF binding site (see *Badis et al., 2009*; *Rohs et al., 2010*; *Jolma et al., 2013*) for a TF that has a stronger primary binding motif. Of the secondary models, 29 cases were due to strong dinucleotide preferences, 21 cases were due to flanking sequence variants, and 40 cases were due to spacing and orientation preferences (*Figure 1D*). Seed sequences and SELEX cycles used are indicated in *Supplementary file 1D*, and the obtained binding profile logos are in *Figure 1—figure supplement 2-13* and all quantitative PWMs are in *Supplementary file 1E* and *Supplementary file 2*.

## Amino-acid sequence similarity predicts TF DNA-binding specificity

As suggested by previous studies (*Wei et al., 2010*; *Jolma et al., 2013*; *Weirauch et al., 2014*), different TF structural families had clearly divergent specificities, and amino-acid sequence similarity was predictive of overall DNA binding specificity in most TF families. For example, we could classify fruit fly bHLH proteins into six subfamilies based on their preferred 6 bp core motifs (*Figure 2*). Members that are similar in amino-acid sequence generally recognized the same core sequence, and highly similar TFs recognized also similar flanking nucleotides (e.g., HLHm3, HLHmbeta, HLHmgamma, HLHmdelta). Order based on similarity of TF DNA binding specificities, calculated using correlations between scores of all possible gapped 6 bp subsequences (see 'Materials and methods' for details), largely recapitulated the phylogenetic relationships between the DBDs (*Figure 2—figure supplement 1*).

## Comparison with existing *Drosophila* motifs

Several previous studies have analyzed TF DNA binding specificities in *Drosophila* using methods different to those used here. The highest coverage fruit fly TF-specificity dataset has been generated using the bacterial one hybrid (B1H) and DNase I footprinting methods (*Noyes et al., 2008*; *Zhu et al., 2011*; *Enuameh et al., 2013*). This collection contains a partial or complete specificities for a total of 325 fruit fly TFs. The profiles we obtained for the proteins that had been analyzed using B1H showed, with very few exceptions, the same core binding specificities (*Figure 3—figure supplement 1*).

To compare the similarity of the datasets in detail, we used the motif similarity score from SSTAT (*Pape et al., 2008*) to draw a dendrogram (*Figure 3—figure supplement 1*). Motifs generally clustered based on the corresponding TF families. However, motifs obtained using HT-SELEX and B1H for the same proteins were often not located next to each other. One reason for this is that flanking sequences obtained using HT-SELEX and B1H are often not identical. The HT-SELEX data also generally yield somewhat longer profiles (average width of 12.7 bp vs 10.7 bp), particularly in cases where TFs bind as dimers (e.g., nuclear receptors). We and others have shown earlier that inclusion of dimeric sites and the extended flanking specificity improve prediction of in vivo occupied sites (*Jolma et al., 2013*; *Orenstein and Shamir, 2014*), indicating that the extended specificity information revealed by HT-SELEX is biologically relevant.

In addition, the HT-SELEX data include binding models for 65 genes, which are not included in the B1H collection (*Supplementary file 1C*). In conclusion, our dataset and the existing B1H dataset support each other, but appear to have method-specific differences and do not overlap completely.

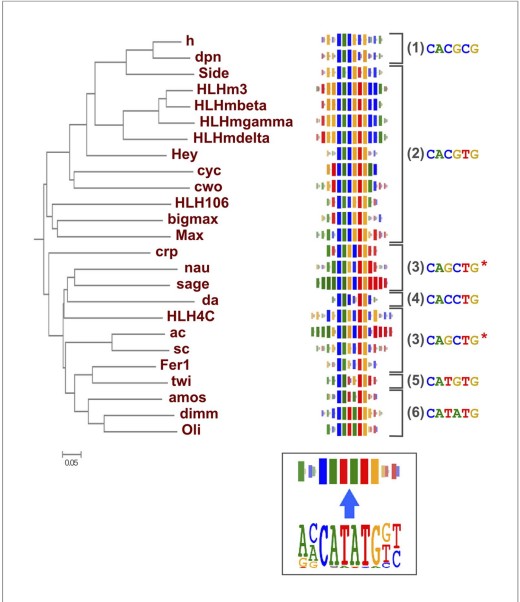

**Figure 2**. Relationship between similarities of TF DBD amino-acid sequence and binding specificity. Barcode logos (middle) for bHLH DBDs arranged according to the amino-acid sequence similarity indicate that sequence conservation is predictive of binding specificity. Inset shows an example of a conversion of a sequence logo into a Barcode logo. For each position, the frequency of each base is indicated by the width of the corresponding colored bar; the intensity of color and the height of the bars at each position are determined by the base with the highest frequency. The core recognition sequences are also indicated (right). Note that structurally close TFs recognize the same core sequences and similar flanking sequences (e.g., HLHm3, HLHmgamma, and HLHmdelta). Note also that CAGCTG motif is recognized by two distinct clades (Asterisks). For analysis of other TF families, see *Figure 2—figure supplement 1*.

The following source data and figure supplements are available for figure 2:

**Source data 1**. Raw Data for 2D heatmap.

**Figure supplement 1**. Heatmap showing similarity of binding profiles and amino-acid sequence similarity score (blastp) between all TFs studied.

## Conservation of primary DNA-binding specificities between *Drosophila* and humans

TFs are generally highly conserved in evolution. However, sequence analysis of DBD sequences of *Drosophila* and human revealed substantial differences in conservation of orthologous TF DBD pairs both between and within different TF families (*Figure 3—figure supplement 2*), suggesting that some divergence of TF binding specificity could have occurred. To address the conservation of the binding specificities, we first calculated motif similarity score of all previously existing B1H and HT-SELEX binding profiles for *Drosophila* and mammals, respectively, and illustrated their similarities as a network map (*Figure 3*). Analysis of similarity between canonical homeodomain TF binding specificities between fruit fly and mammals using these existing datasets suggested that fruit fly homeodomains bind to a smaller subset of sites than human homeodomains (*Figure 3A*). However, this result could be due to the method-specific differences discussed above rather than true evolutionary divergence. Indeed, comparison between our HT-SELEX derived fruit fly and mammalian TF profiles indicate that fruit fly and human homeodomains bind to a similar range of sequences (*Figure 3B*). Analyses of other families, including T-box proteins, Nkx type homeodomains, and C2H2 zinc fingers of the EGR family using B1H data produce similarly misleading results suggesting divergence of DNA-binding specificity. In contrast, comparison of our results generated using the same method yields very similar specificity profiles between orthologous proteins from human and fruit fly (*Figure 3C–E*). Consistently with this finding, analysis of in vivo motifs for Nkx and EGR proteins supports the motif derived by HT-SELEX (*Figure 3—figure supplement 3*).

In addition, several reports have also described structural differences of TFs between fruit flies and mammals. For example, bZIP proteins of the CREB3 subfamily were classified into three subclasses A, B, and C based on the presence or absence of TM and MD1 domains. Fruit fly CrebA is subclass C, whereas its human orthologs CREB3L1 and L2 belong to subclass A (*Barbosa et al., 2013*). However, no differences between DNA binding specificities of these proteins were revealed by our analysis. Likewise, we did not observe differences between orthologous proteins in AP2, bHLH, GCM, HMG, MADS, RFX, and T-box families (*Figure 1—figure supplement 2–13* and not shown).

A global comparison between fruit fly and mammalian TF binding specificities also revealed that fruit fly DNA-binding specificities cover almost the complete range of mammalian TF specificities (*Figure 4*), and that clustering of motifs is not influenced by the species indicating

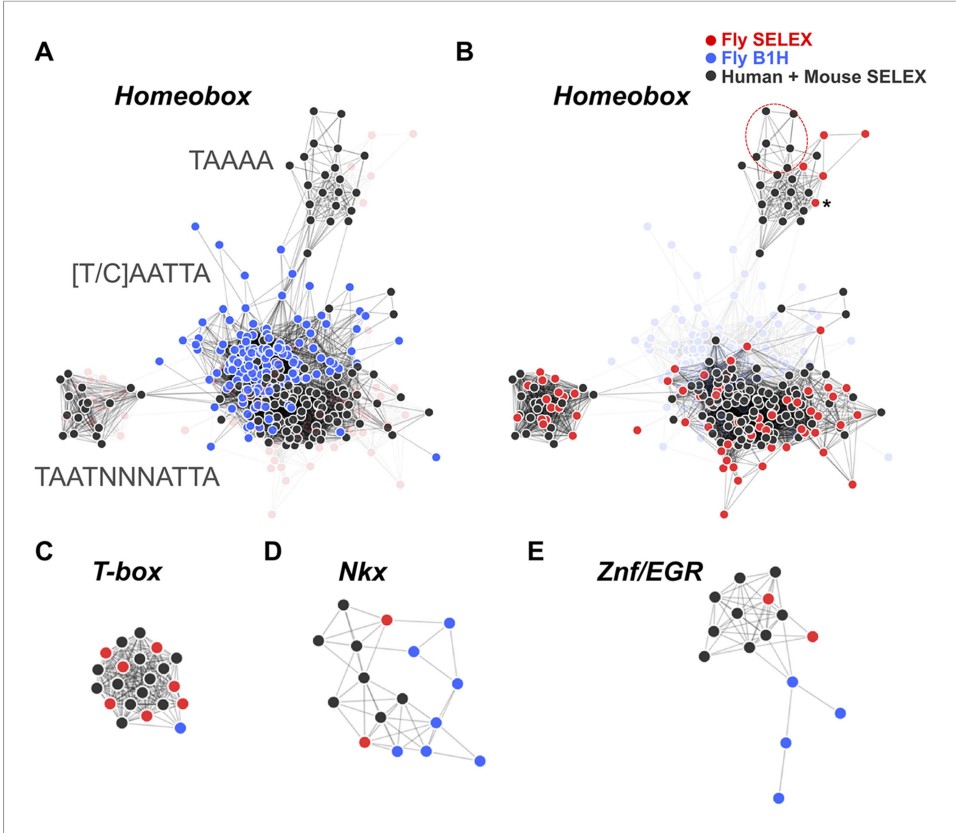

**Figure 3**. HT-SELEX reveals similarity of binding profiles between fruit flies and humans. (**A**) Network graph of previously determined mammalian and fruit fly homeodomain protein DNA binding specificities. Mammalian and fruit fly TF models are represented by black and blue nodes, respectively, and an edge is drawn between similar models. Note that based on existing data, it appears that fruit fly homeodomains (blue) recognize only a subset of the mammalian homeodomain specificities. Partial consensus sequences for the node clusters are also indicated. (**B**) Increased precision obtained by using HT-SELEX for both fruit fly and mammalian TFs reveals that the range of mammalian homeodomain specificities is covered by fruit fly TFs. Note also that *Drosophila* has only one posterior homeodomain protein (Abd-B) that recognizes a motif (asterisk) that is similar to motifs bound by human HOX9-12. *Drosophila* thus lacks a protein whose motif preference is similar to that of human HOX13 proteins (six models inside red oval; see also *Jolma et al., 2013*). (**C**, **D**, **E**) Analysis of previous models for T-box, Nkx homeodomain and EGR proteins. Note that previous data for mammals (black) and fruit flies (blue) suggest divergence of the fruit fly specificity, whereas the fruit fly HT-SELEX data (red) reveal the similarity of the mammalian and fruit fly binding profiles. See *Figure 3—source data 1* for details. See also *Figure 3—figure supplements 1–3*.

The following source data and figure supplements are available for figure 3:

**Source data 1**. Raw Data and a script for the Motif network construction.

**Figure supplement 1**. Similarity of binding profiles generated using HT-SELEX and B1H.

**Figure supplement 2**. Comparison of amino-acid sequence similarity score (from blastp) of indicated protein domains of *Drosophila* and human ortholog pairs.

**Figure supplement 3**. Comparison of motifs defined by HT-SELEX and B1H.

that orthologous proteins recognize highly conserved motifs. These results indicate that the gene regulatory code that determines how DNA sequence directs gene expression is highly conserved in evolution.

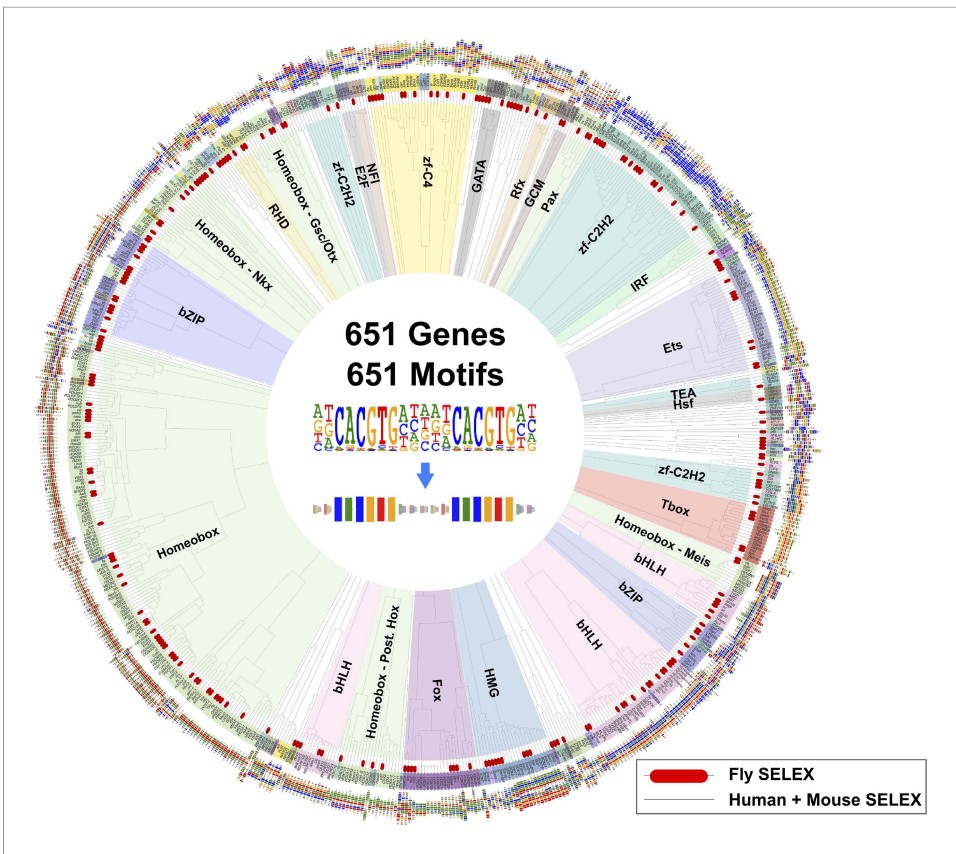

**Figure 4**. Similarity of primary binding profiles between fruit fly and human + mouse. Dendrogram shows motif similarities between the fruit fly motif collection in this study and the human and mouse HT-SELEX collection of (*Jolma et al., 2010*; *Jolma et al., 2013*). Where both human and mouse motifs exist, human motif is shown. *Drosophila* models are indicated by red bars. Barcode logos for each factor are also shown. An example of conversion of a sequence logo into a Barcode logo is shown in center. See *Figure 4—source data 1* for details.

The following source data is available for figure 4:

**Source data 1**. Raw Data and a script for the Dendrogram construction.

## Conservation of secondary binding modes and dinucleotide preferences of TFs

Conservation of primary TF binding specificity is perhaps not that surprising, given the structural constraints of TF folds, and the fact that in many cases a large number of target sites would need to co-evolve with the TF binding specificity. However, more subtle changes in TF DNA-binding specificity could occur, resulting in morphological changes due to loss or gain of secondary TF-DNA binding modes, or due to slow alteration of TF specificity.

To determine whether secondary TF-DNA binding modes are conserved, we compared secondary modes of binding between TFs in several structural families. In all cases where sufficiently high quality data existed for detection of weaker binding sites in both species, the secondary modes were conserved (*Figure 5A* and not shown), indicating that TF specificity does not commonly evolve by loss or gain of secondary binding modes.

To detect more subtle drift in TF specificity, we analyzed conservation of dinucleotide preferences resulting from structure-based TF-DNA recognition (*Rohs et al., 2010*; *Jolma et al., 2013*). To rapidly analyze the dinucleotide preferences, we developed a tool to visualize the extent to which a model based on independence of binding of individual bases (PWM) fails to describe the observed distribution of nucleotide pairs at all base positions. The score is based on the fraction of counts that

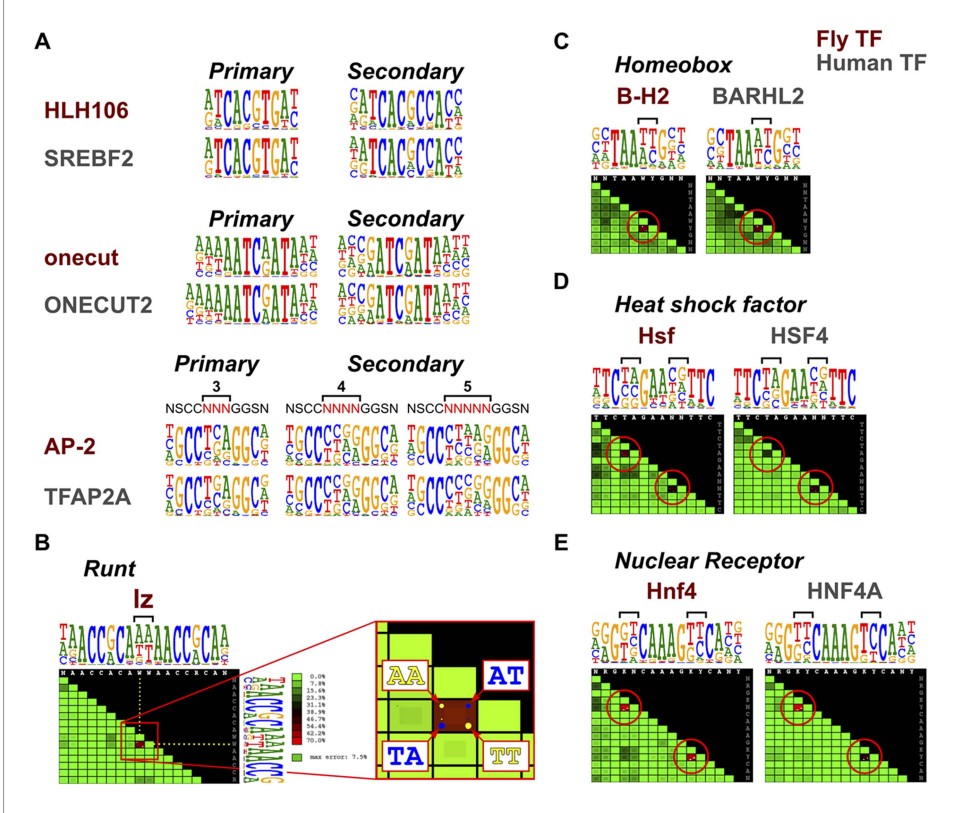

**Figure 5**. Conservation of TF secondary binding modes and dinucleotide preferences. (**A**) Conservation of secondary binding modes. Sequence logos showing primary and secondary binding specificities for the indicated *Drosophila* (red typeface) and human (gray typeface) transcription factors are shown. Note that similar secondary modes exist for all factors. (**B**) Heatmap showing interdependency between bases in the binding model of the Runt family TF lz. Color of each tile indicates the deviation of the observed base distribution from a prediction using a mononucleotide (PWM) model that assumes independence of the indicated bases (color scale on the right; red indicates high deviation). Bracket indicates the two bases that show the largest deviation. Inset shows magnification of the tile; dots inside the each tile indicate pairs of bases that are over- (yellow) or under- (blue) represented relative to mononucleotide model prediction. (**C**, **D**, **E**) Conservation of dinucleotide preferences of the indicated orthologous proteins from fruit fly and human. Base positions deviating from mononucleotide model are indicated by red circles on the heatmap and brackets above the sequence logos.

are mispredicted, being maximal in cases where only two dinucleotides (e.g., AA and TT) are allowed and present at an equal frequency. In such a case, a mononucleotide model mispredicts half of all of counts to the dinucleotides AT and TA that are not bound at all. An almost as extreme case of dinucleotide preference is observed in a homodimeric site of the Runt family gene, lozenge (lz) (*Figure 5B*).

Analysis of conservation of such dinucleotide preferences in several TF families, including homeobox, heat shock factor, and nuclear receptors indicated that in all cases, even the dinucleotide preferences of the TFs were conserved between fruit flies and humans (*Figure 5C–E*). Taken together, these results indicate that orthologous TFs bind to DNA in a very similar manner, resulting in conservation of even the secondary DNA binding modes and subtle dinucleotide preferences.

## Evolution of TF binding specificity via gene duplication followed by divergence

TFs are generally highly conserved between fruit flies and humans (*Choo and Russell, 2011*). However, both fruit flies and humans have a TF class that recognizes a unique site and does not exist in the other lineage, brinker and interferon regulatory factor (IRF) TFs, respectively. In addition, there

are several subfamilies of TFs that recognize unique sites and are species-specific. For example, our analysis indicates that fruit fly lacks monomeric TFs that recognize sites similar to those of non-canonical E2F repressors (E2F7 and E2F8; see *Jolma et al., 2013*) and members of the class III ETS family (*Wei et al., 2010*) (*Figure 1—figure supplement 4* and data not shown). In addition, *Drosophila* has only one protein (ERR) that is homologous to human nuclear receptor group that includes three subgroups, androgen receptor (AR), estrogen receptor (ESR), and estrogen-receptor related protein (ESRR) subgroups. The fruit fly ERR protein recognizes a site that is similar to that recognized by human ESRR. However, no fruit fly orthologs exist for any of the proteins in the ESR or AR subgroups, which recognize different sites (*Figure 6A*). Conversely, the fruit fly knirps-like (knrl) nuclear receptor (NR0 subfamily) has no orthologs in human (*King-Jones and Thummel, 2005*) and its motif is distinct from the other NR proteins (*Figure 1—figure supplement 9*). Similarly, the Pax family transcription factor, Poxn, has no ortholog in human (*Hill et al., 2010*). While other Pax TFs recognize a monomeric Paired binding site, Poxn binds to a dimeric variant of the site (*Figure 6B*). These results indicate that TF DNA-binding specificity can evolve via duplication and divergence, and that such changes can yield novel DNA-binding specificities.

## Changes in binding specificity between orthologs

Analysis of all our data also resulted in the identification of two cases where a direct ortholog between fruit fly and humans had altered specificity. CG30420 (also known as Atf-2) is an ortholog of mammalian ATF2 and ATF7 (*Sano et al., 2005*; *Figure 7A*). This gene recognizes a GACGT(C/G) sequence, whereas all other Atf subfamily genes including ATF7 and ATF2 recognize GACGT(C/A) (*Figure 7B* and *Figure 1—figure supplement 4*). Analysis of enrichment of 10 bp subsequences (10-mers) indicates that one set of 10-mers containing GACGT(C/A) are enriched by both CG30420 and ATF7, whereas 10-mers containing GACGTG sequences are enriched only by CG30420 (*Figure 7C*), indicating that CG30420 can recognize a larger variety of sequences than ATF2 or ATF7. The specificity of CG30420 was observed for both DBD and full-length TF, and also reproducible using a completely independent DBD expression construct. Consistently with the change in specificity, a key asparagine that hydrogen bonds to the cytosine base in other CRE/ATF bZIP proteins is replaced by an arginine in CG30420 (*Figure 7—figure supplement 1*).

Similarly, Hr96, a *Drosophila* NR1J group nuclear receptor gene recognizes tail-to-tail repetitive GT (A/G)CA motifs while the related human NR1 genes recognize head-to-tail repetitive GTTCA motifs (*Figure 7D,E*). Likewise, the class IV ETS family protein SPDEF and its ortholog Ets98B recognized the same core sequence, but differed in their preferences for 3′ flanking sequences (*Figure 1—figure supplement 4*). Taken together, these results indicate that in some isolated cases, specificities of orthologous TFs can diverge over long evolutionary time. Whether the divergence has originally occurred via duplication and divergence, followed by loss of different paralogs in each lineage is not clear.

## Discussion

### High resolution binding profiles of *Drosophila* TFs

We report here high-resolution DNA-binding specificity for a large fraction of *Drosophila* TFs. The binding models were determined using HT-SELEX, the same method we have used previously to determine specificities of mammalian TFs. Consistent with our earlier data, we found that TFs and TF DBDs recognize the same, relatively long sites, with more than half of all models being >10 bp in length (*Jolma et al., 2013*). To analyze the data, we developed novel computational algorithms, including a novel distance metric for gapped subsequences, score for dinucleotide preferences, and a method for comparing TF binding specificity models to each other. In addition, we developed a new barcode logo that facilitates visualization of similarities and differences in PWMs. These methods will have many applications in DNA sequence-analyses that extend beyond analysis of TF binding specificity.

The high resolution of the data coupled with the novel computational and data visualization tools allowed comparison of TF binding specificities at an unprecedented scale and resolution, revealing a striking level of conservation of binding specificities that extends to homodimeric sites, minor secondary binding modes, and subtle dinucleotide preferences. This result is surprising even considering the similarity of protein sequences between DBD ortholog pairs, and also in marked contrast to the reported low degree of conservation of protein–protein interactions between species

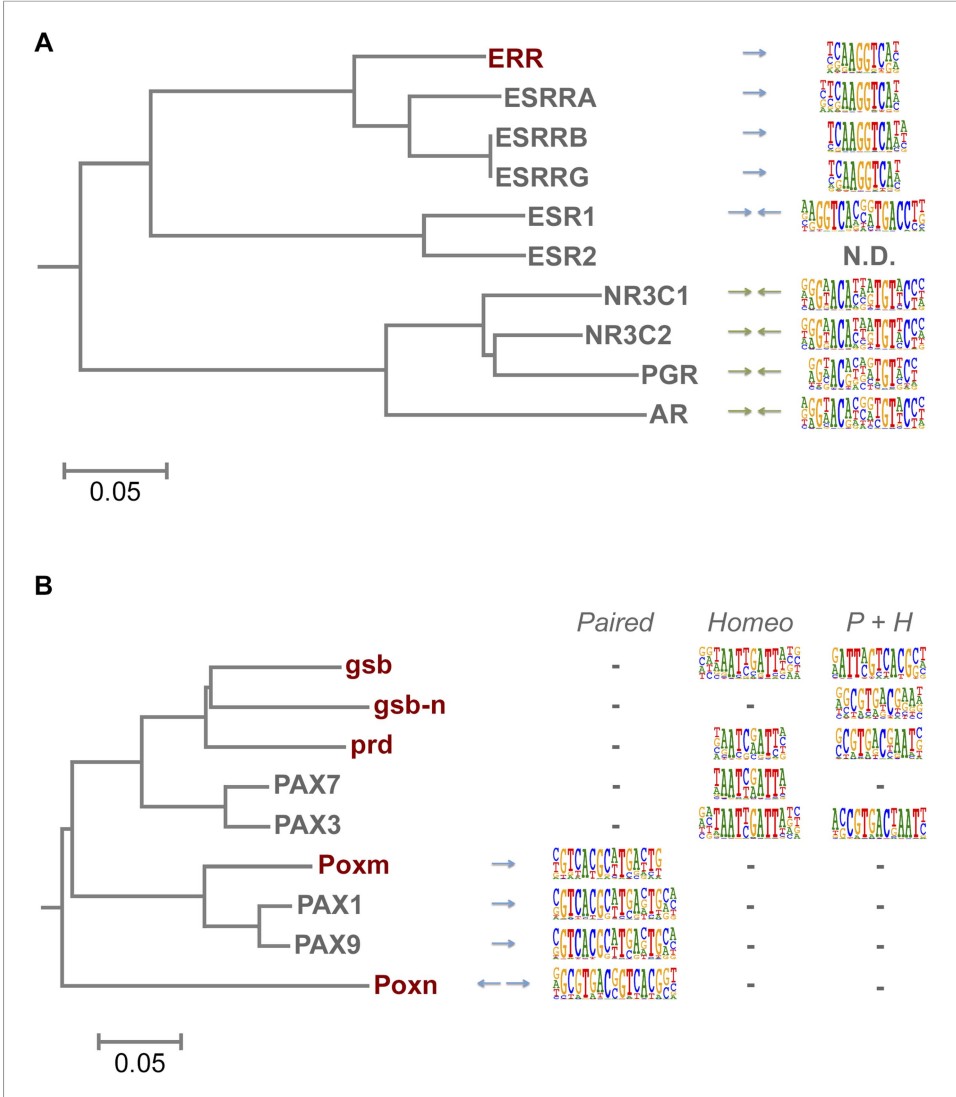

**Figure 6**. Evolution of TF binding specificity by gene duplication and divergence. (**A**) Duplication and divergence has generated three distinct specificities in related nuclear receptors. Dendrogram is drawn based on amino-acid sequence similarity between the DBDs. Binding motifs are shown on the right column. Only one of the specificities (top clade), a monomeric AAGGTCA motif (blue arrow) exists in *Drosophila*. A human-specific site (middle clade) recognized by ESR1 is a dimer with the same half-site, but in a head-to-head configuration. Another human-specific site (bottom clade) recognized by androgen receptor has a different half-site (G/A)G(A/T)ACA (green arrow).
(**B**) Specificity of the Pax proteins in relation to the sequence similarity of their paired DBDs. Binding motifs (right) are categorized into paired, homeodomain (homeo), and paired and homeodomain (P + H). Note that fruit fly Poxn binds to a site that is not recognized by any of the human PAX proteins. Arrows indicate an orientation of the paired motifs. N.D. = not determined. Complete data for all major families are shown in *Figure 1—figure supplement 2–13*.

(*Gandhi et al., 2006*). Despite significant number of differences in the number of TFs, most binding specificities present in mammals were also identified in *Drosophila*, suggesting that the gene regulatory code is highly conserved.

## Comparison with earlier data

Previously, Wolfe and colleagues have reported a dataset of *Drosophila* TF motifs (*Noyes et al., 2008*; *Zhu et al., 2011*; *Enuameh et al., 2013*). Much of these data are obtained using B1H analysis.

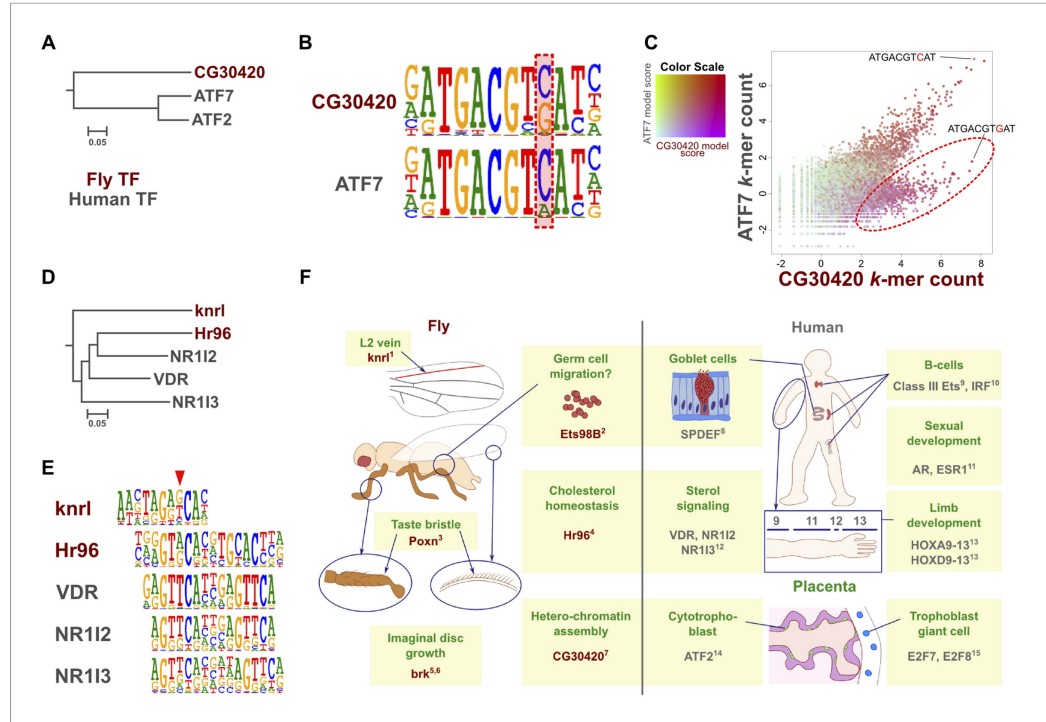

**Figure 7**. Evolution of TF binding specificity by duplication and divergence. (**A**, **B**, **C**) The *Drosophila* bZIP protein CG30420 recognizes a site that is different from those recognized by its human orthologs ATF2 and ATF7. Box in B indicates the position whose specificity is diverged between fruit fly and human. Note that CG30420 recognizes a 10-mer ATGACGT<u>G</u>AT that is not bound by ATF7. Enrichment of 10-mers in CG30420 and ATF7 experiments is shown in panel C, oval indicates 10-mers preferentially recognized by CG30420. Two-dimensional color scale indicates the score of each *k*-mer against CG30420 and ATF7 PWM models (red indicates strong match to both, violet and green strong matches to only CG30420 and ATF7, respectively). For replicates and structural analysis, see *Figure 7—figure supplement 1*. (**D**, **E**) The diverged binding specificities between *Drosophila* and human in nuclear receptor subfamily. Whereas, all human NR1I subfamily genes (VDR, NR1I2, NR1I3) recognize a motif containing direct repeat of a GTTCA motif, Hr96 (NR1J subfamily) recognizes tail-to-tail dimer of a different half-site, GT(G/T) CA. *Drosophila* knrl (NR0 subfamily), which has no human ortholog, recognizes a G(A/G)(G/T)CA motif, which is not recognized by any human nuclear receptor. The diverged position in NR1 subfamily is indicated by red triangle. Dendrograms in A and D show amino-acid sequence similarity of the DNA-binding domains. (**F**) Summary of biological roles of TFs with divergent specificities. Cell types and biological functions are indicated in green in *Drosophila* (left) and human (right). For some TFs with multiple functions (e.g., bZIP and HOX proteins, nuclear receptors), only one divergent role is shown for clarity. In addition to their divergent roles, Hr96 and its orthologs have also shared functionality (xenobiotic responses; (*King-Jones et al., 2006*; *Reschly and Krasowski, 2006*)). Note that TFs with novel specificities are often associated with cell types that do not exist in the other organism. References: [1](*Lunde et al., 1998*); [2](*Hsouna et al., 2004*); [3](*Boll and Noll, 2002*); [4](*Horner et al., 2009*); [5](*Schwank et al., 2008*); [6](*Doumpas et al., 2013*); [7](*Seong et al., 2011*); [8](*Gregorieff et al., 2009*); [9](*Bartel et al., 2000*); [10](*Lu et al., 2003*); [11](*Oliveira et al., 2004*); [12](*Pardee et al., 2011*); [13](*Zakany and Duboule, 2007*); [14](*Maekawa et al., 1999*); [15](*Chen et al., 2012*). Raw data and scripts are provided at *Figure 7—source data 1*.

The following source data and figure supplements are available for figure 7:

**Source data 1**. Raw Data for kmer plot.

**Figure supplement 1**. Validation of the difference in specificity between CG30420 and CRE family bZIP proteins.

Our data generally show good agreement with these data at the level of consensus and/or core TF binding sequences. However, the flanking sequences recognized by many TFs (*Jolma et al., 2013*) are poorly defined in the B1H data, probably at least in part due to the fact that a large fraction of the B1H data is derived from relatively few sequence reads generated using Sanger-sequencing. In addition,

our data are generated using more than 10,000-fold more complex library, allowing analysis of longer dimeric motifs and more complex binding modes (e.g., secondary motifs and dinucleotide preferences). Thus, in addition to extending the repertoire of *Drosophila* TF binding specificities by providing 122 novel high-resolution motifs, our dataset extends previous models by shedding light on new features of *Drosophila* TF DNA-binding characteristics.

## Conservation of transcription factor binding specificity between fruit flies and mammals

TFs that belong to the same structural family generally bind to similar sites (e.g., *Figure 2—figure supplement 1* and *Jolma et al., 2013*), which is thought to represent the limited sequence space that can be recognized by particular protein folds. We show here that structural similarity of proteins defines not only the similarity of the core DNA binding site but also the flanking sequences, and that almost all specificities available for a human are also present in fruit flies (*Figures 2, 4*). *Drosophila* has many genes that are orthologous to human TF genes with highly conserved DBD protein sequences even though these species diverged over 600 million years ago (*Figure 4*; *Figure 3—figure supplement 2*). We show here that orthologous TFs have almost identical core DNA binding profiles, flanking sequences, and secondary modes of binding (*Figure 5* and *Figure 1—figure supplement 2–13*). This fact suggests that DNA binding specificity is highly constrained by protein structure and is difficult to alter even over long evolutionary time scales. These results also indicate that TF specificity is not generally subject to subtle drift, but tends to either stay very similar or evolve to a distinctly different state.

## Divergence of specificity

Although humans have a larger number of TFs than fruit flies, the gene duplication events that have occurred after separation of the invertebrate and vertebrate lineages have generally not lead to emergence of novel TF specificities, outside of the rapidly evolving human zinc finger proteins that suppress endogenous retroviruses (*Rowe et al., 2010*; *Lukic et al., 2014*). However, we did find a few cases where a clear divergence of TF specificity had occurred. Most cases involved TFs whose genes have duplicated, and then diverged. In several cases, orthologs of a human protein with a particular specificity do not exist in fruit flies. Examples include non-canonical E2F repressors (E2F7, E2F8), ETS class III (SPI1, SPIB, and SPIC), and nuclear receptors of the ESR and AR subgroups. Conversely, the fruit fly nuclear receptor knirps-like (knrl) and the Pax protein Poxn do not have direct human orthologs, and recognize sites not bound by human TFs. In addition, the arthropod specific TF, brk, recognizes a unique binding site.

We also identified few cases where an apparent direct ortholog has acquired a novel specificity in either vertebrates or invertebrates (*Figure 7*). One case affected the bZIP proteins CG30420 and its human orthologs ATF2 and ATF7, and another the nuclear receptors Hr96 and NR1I. In addition, the human ETS class IV proteins, SPDEF, and its fruit fly ortholog Ets98B recognized an identical GGAT core motif, but preferred different sequences in their 3′ flanks. Surprisingly, recognition of a secondary site we have previously described for SPDEF (*Jolma et al., 2013*) was conserved in fruit flies, with nearly identical sites bound by both SPDEF and Ets98B. The conservation of the secondary binding mode despite divergence of the primary specificity indicates that the binding modes can in some cases evolve independently of each other. Protein sequence-level conservation of the DNA-binding domains of the ETS proteins suggests that the divergence in specificity occurred at or near the divergence of invertebrates and vertebrates (not shown).

## Biological roles of TFs with divergent specificity

Individual cell types of a multicellular organism are defined by expression of specific combinations of TFs (*Takahashi and Yamanaka, 2006*), and the number of TF genes correlates with complexity of organisms (see e g., *Mastracci and Sussel, 2012*). Interestingly, many of the TFs that we find to have a different specificity in humans are related to endocrine system function (AR, ESR1), or physiology of cell types that do not exist in fruit flies (*Figure 7F*). For example, ATF2 and non-canonical E2F repressors are important for placental morphogenesis (*Chen et al., 2012*; *Maekawa et al., 1999*), whereas ETS class III and IRF factors have important roles in differentiation of cells of the adaptive immune system (*Bartel et al., 2000*; *Lu et al., 2003*). SPDEF, in turn, regulates goblet cell differentiation in lung, conjunctiva,

and intestine (*Chen et al., 2009*; *Gregorieff et al., 2009*; *Marko et al., 2013*). Although *Drosophila* has intestinal stem cells and enterocyte-related cells, it does not have a cell type similar to mammalian goblet cells (*Jiang and Edgar, 2012*). These results suggest that evolution of novel TF binding specificities has contributed to emergence of novel types of differentiated cells.

Taken together, our results indicate that human and fruit fly TF binding specificities display a striking level of conservation, despite dramatic morphological differences resulting from more than 600 million years of evolution and lack of detectable sequence conservation at the level of *cis*-regulatory elements. These results indicate that analogously to the genetic code, which is more conserved than protein-coding DNA (*Moura et al., 2010*), the gene regulatory code is much more conserved than the regulatory sequences themselves. Our results suggest that morphological divergence is not driven by subtle drift in specificity of TFs, but primarily caused by *cis*-regulatory changes (*Carroll, 2000*; *Levine and Tjian, 2003*), with some contribution from relatively large shifts in binding specificities of specific TFs.

## Materials and methods

### Data availability
HT-SELEX sequence data are deposited to European Nucleotide Archive (ENA, EMBL-EBI) under accession PRJEB7373.

### Clone collection and protein production
Collection consisting of 760 DNA-binding domains was cloned by PCR, or generated by gene synthesis (codon optimized, Genscript) as indicated in *Supplementary file 1A*. DBD sequences for the collection were selected using Pfam or Interpro predicted DBDs, with addition of 4–5 flanking amino acid residues. Collection of 633 fruit fly full-length TF clones has been previously described (*Hens et al., 2011*). Constructs were verified by sequencing at least one end using a capillary sequencer (MWG, Germany). Clones were transferred to N-terminal thioredoxin hexahistidine bacterial expression vector (pETG-20A; *Vincentelli et al., 2011*) by a Gateway LR reaction (Invitrogen, Carlsbad, CA). The Rosetta (DE3)pLysS (Novagen) strain was used for protein production. *E. coli* cells were cultured in two wells of a deep 96-well plate (Thermo, AB0661) at 17–20°C in ZYP5052 autoinduction medium (see *Vincentelli et al., 2011*). Expressed proteins were purified using His-tag with Ni Sepharose 6 Fast Flow resin (GE Healthcare) and a 20 μm pore size filter plate (Nunc, 278011) as described in *Vincentelli et al. (2011)* and *Jolma et al. (2013)*. All proteins except NFI were expressed in *E. coli*. NFI was expressed in *Drosophila* S2 Schneider cells using the expression vector pMT-Dest-HisSBP3xV5 (*Bonke et al., 2013*), protein expression was performed as described previously (*Bonke et al., 2013*).

### Identification of secondary binding modes and dinucleotide preferences
Similarity of gapped and ungapped subsequences were analyzed using a novel distance metric 'Huddinge distance', that is based on number of defined and alignable bases between the compared subsequences. Formally, Huddinge distance between two gapped or ungapped subsequences is defined as $d - a$, where $d$ is the maximum number of defined bases in either of the compared subsequences, and $a$ is the maximum number of bases that can be perfectly aligned between them without introduction of new gaps. Subsequences that had higher counts than any of their neighbors at Huddinge distance of one were identified as local maxima, and used as initial seeds for the generation of the binding profiles.

The Autoseed method is in principle capable of identifying any specificity that results in enrichment of subsequences longer than 2 bp. In general, longer subsequences are expected to be more rare and thus more easily identified. Thus, HT-SELEX combined with Autoseed may not be able to identify specificity for relatively non-specific or accessory DNA-binding proteins that display weak mono-nucleotide or dinucleotide preferences. Scripts and software are supplied as Supplementary files.

### HT-SELEX and PWM models
HT-SELEX was performed essentially as described in *Jolma et al. (2013)* using selection ligands containing a 6 and 2–3 bp barcode before and after the 20 or 40 bp randomized region, respectively. The experimental reproducibility using 20 and 40 bp randomized regions is shown in *Figure 1—figure supplement 14*. Sequences of selection ligands are indicated in *Supplementary file 1B*. The DNA ligands were mixed with purified proteins in a binding buffer (10 mM Tris–HCl

(pH 7.5), 50 mM NaCl, 1 mM MgCl$_2$, 0.5 mM DTT, 0.5 mM EDTA, 4% glycerol, 5 µg/ml poly (dI-dC)) or a crude lysate of NFI expression vector transfected S2 cells, and were incubated at room temperature for 10 min. Then Ni Sepharose 6 Fast Flow resin (GE Healthcare), or High capacity Streptavidin Agarose resin (Thermo, 20,359) for NFI, equilibrated in binding buffer was added and the mixture was incubated for additional 20 min. After washing 12 times to remove nonspecifically bound oligonucleotides, the complexes were suspended in milliQ water. Subsequently, bound selection ligands were amplified by PCR using Phusion DNA polymerase (Finnzymes), and the resulting products used as selection ligands for the next round of selection. This process was repeated four to seven times. After each cycle, the selection ligands were pooled and sequenced using Illumina HiSeq 2000 sequencer. Raw sequencing data were binned according to barcodes and used for further analyses.

PWM models were generated using initial seeds identified using IniMotif (*Jolma et al., 2010*) and/or Autoseed (above) that were refined by expert analysis as described in *Jolma et al. (2013)*. Exact seeds, cycles, and multinomial model used are indicated in *Supplementary file 1D*. For comparison with *Drosophila* models in *Figures 6 and 7*, Human HT-SELEX data were generated for PAX3, PGR, NR1I2, and NR1I3 using *E. coli* expressed DBDs, and the obtained data and data from *Jolma et al. (2013)* were reanalyzed using the Autoseed-based pipeline.

## Data visualization

All dendrograms based on amino-acid sequence similarity were generated by PRANK (Probabilistic Alignment Kit) v. 121218 (*Löytynoja and Goldman, 2005*) using default parameters and DBD sequences predicted by Pfam or Interpro.

Barcode logos were designed to allow visual alignment of sequences based on similarity of color profiles. At each base position, four bars are drawn that represent the frequency of the bases. Width of each bar is proportional to the frequency of the corresponding base (range 0–1), and height and color intensity of all the bars at a given position are proportional to the frequency of the most common base at that position (range 0.25–1).

To illustrate the correlation between protein sequence similarity and motif similarity (*Figure 2—figure supplement 1*), we drew a two dimensional heatmap. Amino-acid similarity scores were calculated using Blastp (*Altschul et al., 1990*) with same DBD sequences as for the phylogenetic trees. The score is calculated by multiplying the Blastp score by matching length, and then dividing it by the length of the shorter query sequence. Where more than two DBDs were present in the compared TFs (e.g., Pax family), the score from the comparison resulting in the higher score was used.

To draw the box plot for the structural domain sequence conservation (*Figure 3—figure supplement 2*), the orthologous gene list was downloaded from Ensembl biomart (version 78; genome assembly GRCh38). In case one fruit fly gene had more than one human ortholog, the human gene with the highest score was selected for the analysis. Protein conservation score was calculated by blastp as described above. Where more than one DBDs were present in the TFs analyzed (e.g., Pax), the protein sequences of all the DBDs were concatenated to a single sequence.

Motif network graph (*Figure 3*) and dendrogram (*Figure 4*) are based on motif distance scores calculated by SSTAT (*Pape et al., 2008*). Motif similarity network (see *Jolma et al., 2013*)) based on data generated in this work, and on existing B1H data (flyfactorsurvey database, downloaded 19 Feb 2013) and HT-SELEX data from *Jolma et al. (2013)* was drawn using Cytoscape V3 (*Lopes et al., 2010*). Motif dendrogram was drawn by using euclidean distance metric with average linkage with R package ape.

Dinucleotide preference heatmap (*Figure 5*) illustrates the fraction of counts that are mispredicted if a mononucleotide model is used to predict the number of nucleotide pairs at a given pair of base positions. Score is scaled between 0% (bases bind independently of each other) and 100% (where half of all counts are mispredicted).

To compare enrichment of subsequences in two different datasets, scatter-plots of log-odd scores of the subsequence counts were generated using R. To illustrate subsequences that match to PWM models, the subsequences were scored against the indicated PWMs and colored according to the color scales indicated.

Scripts and software are supplied as *Supplementary file 3*.

## Acknowledgements

We thank Bei Wei, and Drs. Martin Enge, Bernhard Schmierer and Minna Taipale for critical review of the manuscript, and Sandra Augsten, Lijuan Hu, Anna Zetterlund, and Sini Miettinen for technical assistance.

## Additional information

### Funding

| Funder | Grant reference number | Author |
| --- | --- | --- |
| Karolinska Institutet | | Jussi Taipale |
| Knut och Alice Wallenbergs Stiftelse | | Jussi Taipale |
| Göran Gustafssons Stiftelse för Naturvetenskaplig och Medicinsk Forskning | | Jussi Taipale |
| Swiss National Science Foundation | CRSI33_127485 | Korneel Hens, Bart Deplancke |
| École Polytechnique Fédérale de Lausanne (EPFL) | Institute Internal Funding | Bart Deplancke |
| Vetenskapsrådet | | Jussi Taipale |

The funders had no role in study design, data collection and interpretation, or the decision to submit the work for publication.

### Author contributions

KRN, Prepared essential clones, Conception and design, Acquisition of data, Analysis and interpretation of data, Drafting or revising the article; AJ, JT, Conception and design, Analysis and interpretation of data, Drafting or revising the article; YY, Acquisition of data, Analysis and interpretation of data; EM, Acquisition of data, Analysis and interpretation of data, Drafting or revising the article; TK, JT, Analysis and interpretation of data, Drafting or revising the article, Contributed unpublished essential data or reagents; JA, KH, Prepared essential clones; BD, Prepared essential clones, Drafting or revising the article; EEMF, Prepared essential clones, Conception and design, Drafting or revising the article

### Author ORCIDs
Korneel Hens, http://orcid.org/0000-0002-0362-7007

## Additional files

### Supplementary files

• Supplementary file 1. (**A**) DBD and full-length clones. (**B**) Selex oligos. (**C**) Summary of HT-SELEX results and comparison to B1H. (**D**) Experimental details, including seed sequence, multinomial level and SELEX cycles used for signal and background. (**E**) *Drosophila* and human PWMs in one table. (**F**) Count of obtained motifs.

• Supplementary file 2. *Drosophila* and human PWMs as separate files.

• Supplementary file 3. Scripts for the total Autoseed, the barcode logo, and the dinucleotide heatmap.

### Major dataset

The following dataset was generated:

| Author(s) | Year | Dataset title | Dataset ID and/or URL | Database, license, and accessibility information |
| --- | --- | --- | --- | --- |
| Nitta KR, Jolma A, Yin Y, Morgunova K, Kivioja T, Akhtar J, Hens K, Toivonen J, Deplancke B, Furlong EEM, Taipale J | 2015 | Binding specificities of Drosophila melanogaster transcription factors | PRJEB7373; http://www.ebi.ac.uk/ena/data/view/PRJEB7373 | Publicly available at the European Nucleotide Archive (ENA, EMBL-EBI) (http://www.ebi.ac.uk/ena/). |

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
