## [Decision Letter]

Thank you for sending your work entitled “Conservation of transcription factor binding specificities across 600 million years of bilatera evolution” for consideration at *eLife*. Your article has been favorably evaluated by Diethard Tautz (Senior editor) and 3 reviewers, one of whom, Bing Ren, is a member of our Board of Reviewing Editors. One of the other two reviewers, Kevin White, has agreed to share his identity.

The Reviewing editor and the other reviewers discussed their comments before we reached this decision, and the Reviewing editor has assembled the following comments to help you prepare a revised submission.

Nitta et al. used HT-SELEX to characterize the DNA binding specificity for close to 900 *Drosophila* transcription factors (TF), generating DNA binding motifs for approximately 230 of them. For many of these TFs, they identified more than one motif recognized by the TFs. The authors then used this tremendous resource to address an important question, are TF binding specificities in *Drosophila* conserved in mammals? To do so, they have developed several new algorithms and generated many high-resolution PWMs for 230 fly TFs. Together with their previous work, this study will have a big impact on the transcription field and data generated will serve as a valuable resources for both experimental and computational researchers. The results not only confirmed previously known *Drosophila* TF binding specificities mapped using different approaches (mainly yeast one hybrid), but also added roughly 110 new TF binding motifs. The key finding is that *Drosophila* and Mammalian TFs with similar protein sequences have highly similar DNA binding specificity, which uncovers a strong conservation of DNA binding specificity of transcription factors over long period of evolution. This is very significant because it argues for a strong negative selection pressure for TF genes. Such conservation, though having been pointed out in previous studies (for example, Noyes MB et al., Cell, 2008), is interesting especially given the large scale and systematic nature. Another contribution is the AutoSeed program, which would be useful for processing of large SELECT datasets for extraction of TF binding motifs. The divergent binding specificity in human is interesting as it might be relevant to the emergence of novel cell types.

Technologically, the experiment was well designed and the analysis was thorough. The manuscript was well written and results were clearly presented.

Main concerns:

1) While the reviewers agree that the high degree of conservation in TF recognition motifs between mammals and flies is a potentially significant finding, to gain novelty some felt that it would be important that the authors, through either new analyses or new experiments, demonstrate the degree to which such conservation is unexpected, or unusual, for example, in comparison with other protein classes.

2) A missed opportunity is that there is relatively little in-depth characterization of TF binding specificity in terms of the relationship between DNA motifs and DNA binding domains for different family of TFs.

3) Since both 20- and 40-mer random sequences were used as DNA ligands, how reproducible were the motifs generated between the two pools of the DNA ligands?

4) Many C2H2 zinc finger and bHLH TFs failed to generate any significant motifs. Did they fail to recognize any 20- or 40-mer sequences or were the sequences they bind to too diverse to generate a significant motif?

5) The authors described comparison between motifs obtained by HT-SELEX and B1H system and concluded that there exists method-specific difference as illustrated in Figures 2 and 3. Have the authors tested the differences using EMSA or cell-based luciferase assays? It is important for the community to fully understand what caused the discrepancy and which method(s) is more reliable.

6) ATFs are known to recognize octameric palindrome sequences. If the authors' discovery of ATGACGTGAT as a new consensus for fly Aft-2 were true, it would disrupt this palindrome. Have the authors applied orthogonal method, such as EMSA and luciferase assays, to validate the expected interactions? Since this is one of the two cases reported in this study that an ortholog TF pair showed divergent binding specificity, the authors should provide more solid evidence.

7) The PWMs in the supplement table is missing. This information should be made publically available.

---

## [Author Response]

In general, the reviewers have a positive response to our manuscript, stating that the manuscript “will have a big impact on the transcription field and data generated will serve as a valuable resources for both experimental and computational researchers”. The reviewers also found that our key finding that “The key finding is that *Drosophila* and Mammalian TFs with similar protein sequences have highly similar DNA binding specificity…” is very significant.”

The reviewers requested additional technical controls, and comparative analyses to analyze the relationship between sequence level and functional conservation. In the revised version, we have addressed all of the concerns of the reviewers.

The changes include rewriting of the text, updates to figure panels in main Figures 1, 2, 3 and 4 and figure supplements Figure 1—figure supplement 2 and Figure 3—figure supplement 1, the addition of five new supplementary figures (Figure 1—figure supplement 3, Figure 2—figure supplement 1, Figure 3—figure supplement 2, Figure 3—figure supplement 3, Figure 7—figure supplement 1), and inclusion of two additional authors, Ekaterina Morgunova, who contributed to the structural analyses, and Jarkko Toivonen, who contributed to the generation of the data for the new analyses.

Main concerns:

*1) While the reviewers agree that the high degree of conservation in TF recognition motifs between mammals and flies is a potentially significant finding, to gain novelty some felt that it would be important that the authors, through either new analyses or new experiments, demonstrate the degree to which such conservation is unexpected, or unusual, for example, in comparison with other protein classes*.

We have now performed extensive analyses of conservation of orthologous pairs of TF DNA-binding domains and other interaction domains involved in protein-RNA, protein-protein and enzyme-substrate interactions. We included the comparative analyses of other interaction domains as previous analyses have revealed poor conservation of protein-protein interactomes between *Drosophila* and human.

This analysis revealed that although many TF DBD families are very highly conserved, several families exist whose conservation level is similar to that of the other analyzed domains (e.g. kinase domains, SH3 and SH2 domains).

In addition, we observed broad differences in sequence-level conservation of DBD ortholog pairs within families. Thus, a simple but reasonable prediction based solely on protein sequence would be that there should be significant divergence in binding specificity between human and *Drosophila* TFs. However, we report here that orthologous pairs of TF DBDs almost invariably bound to the same DNA sequences, indicating that the binding specificity is very deeply conserved. We have now included these new analyses in the manuscript (Figure 3—figure supplement 2), and rewritten it to highlight the fact that the extreme conservation of binding specificity is surprising even after considering primary sequence-level conservation of DBDs (second paragraph of the Discussion, and the subsection headed “Conservation of primary DNA-binding specificities between *Drosophila* and Humans” in the Results).

*2) A missed opportunity is that there is relatively little in-depth characterization of TF binding specificity in terms of the relationship between DNA motifs and DNA binding domains for different family of TFs*.

This is an important point. We have now added a heatmap showing sequence-level conservation and conservation of binding specificity as new Figure 2—figure supplement 1. Raw data is also added as [Supplementary-material SD1-data]. This figure clearly shows that binding specificity is primarily determined by the structural family of the TF. We have also discussed this issue more extensively in the Results section.

3) Since both 20- and 40-mer random sequences were used as DNA ligands, how reproducible were the motifs generated between the two pools of the DNA ligands?

In general, motifs recovered using different length ligands are very similar for all TFs that recognize relatively short (<15 bp) sites. We now show some examples of the reproducibility as new Figure 1—figure supplement 3. We also now discuss this matter in the Methods section and in the legend to Figure 1—figure supplement 3.

4) Many C2H2 zinc finger and bHLH TFs failed to generate any significant motifs. Did they fail to recognize any 20- or 40-mer sequences or were the sequences they bind to too diverse to generate a significant motif?

This is an important point. We repeated the SELEX for the bHLH and ZF proteins, and managed to generate models for four and eight additional proteins, respectively. The low recovery is thus intrinsic to the TFs and not due to low yield of the SELEX experiments themselves.

The low recovery of ZFs motifs is due to several reasons. First, some ZFs may not bind to DNA. Second, many C2H2 zinc finger genes encode large arrays of ZFs, which are difficult to express in *E. coli*.

In the case of bHLH proteins, it is known that some of them lack a functional DBD, and many of them do not bind to DNA as homodimers, and instead function as obligate heterodimers. The orthologs of *Drosophila* bHLH proteins that failed in many cases also failed in human and mouse SELEX analyses (e.g. sim/SIM1, SIM2, Hand/HAND1, HAND2 etc).

Our analysis pipeline is capable of detecting any subsequence that is enriched. A diverse set of subsequences would thus be detected. Thus, it is unlikely that the failure of the experiments was due to lack of our ability to detect enriched features characteristic of TF-DNA binding.

It is, however, possible that generalized and weak preference to mononucleotides or dinucleotides due to shape based DNA recognition would not be detected. We have now included the additional data and discussed the ZF and bHLH cases, and the features of the computational analysis pipeline in the second paragraph of the Results section and in the beginning of the subsection headed “Identification of secondary binding modes and dinucleotide preferences”, also in the Results, respectively.

*5) The authors described comparison between motifs obtained by HT-SELEX and B1H system and concluded that there exists method-specific difference as illustrated in*
Figures 2 and 3*. Have the authors tested the differences using EMSA or cell-based luciferase assays? It is important for the community to fully understand what caused the discrepancy and which method(s) is more reliable*.

In general the HT-SELEX and B1H results are in very good agreement, and mutually support each other. Both sets of data are for most purposes almost equally useful. However, there are minor differences that lead to loss of precision that is necessary for comparison of data for orthologs from different species.

The minor differences are caused mainly by failure of the B1H to capture subtle aspects of TF flanking specificity. In addition, in few cases the B1H analysis reports a monomer when the bound species is a homodimer. Comparison to data obtained using ChIP-seq or DNase I hypersensitivity supports the dimeric species and/or the flanking specificity revealed HT-SELEX where available. The ability of HT-SELEX to capture biologically relevant flanking specificity has also been confirmed by independent groups. For example, when comparing HT-SELEX with protein-binding microarrays, [52] found that: “HT-SELEX models had better [ChIP-seq] binding prediction, partly because of the ability to model the side positions more accurately.” We have also reported similar results (35).

We have now added a Figure 3—figure supplement 3 showing comparisons between some of the most divergent B1H and HT-SELEX models, including those shown in main Figure 3. The models are still similar, but some differences are observed, and where available, data using other methods, including DNase I hypersensitivity from the same lab that performed the B1H supports the HT-SELEX version. We also discuss these issues briefly in the Results section (subsection headed “Conservation of primary 197 DNA-binding specificities between *Drosophila* and Humans”), where we also emphasize that the B1H data is in general of very high quality.

*6) ATFs are known to recognize octameric palindrome sequences. If the authors' discovery of ATGACGTGAT as a new consensus for fly Aft-2 were true, it would disrupt this palindrome. Have the authors applied orthogonal method, such as EMSA and luciferase assays, to validate the expected interactions? Since this is one of the two cases reported in this study that an ortholog TF pair showed divergent binding specificity, the authors should provide more solid evidence*.

There appears to be a misunderstanding here. We have now made it clearer that both fly Atf-2 (CG30420) and ATF7 have the same palindromic consensus 10-mer (ATGACGTCAT). They bind with weaker affinity to other sites, most of which are not palindromic as any single substitution to the site generates a non-palindromic sequence.

It is relatively commonly observed that homodimers bind non-palindromic sites, probably due to interactions between the subunits (see [35] Figure S3A). Also crystal structure data has shown that bZIP TFs bind even to palindromic DNA with asymmetric contacts (Miller et al. 2003, PMID: 12578822 and Schumacher et al. 2000, PMID: 10952992).

We have now included the Results using full-length *Drosophila* CG30420 to Figure 7—figure supplement 1, and performed a replicate experiment using another completely independent CG30420 DBD clone (data also added to the figure supplement). In addition, we have analyzed the primary sequence of CG30420, which reveals a substitution by arginine of a key asparagine that recognizes the C base where the specificity difference is observed. We have now briefly discussed this issue in the Results section (subsection headed “Changes in binding specificity between orthologs”), and added the structure-based analysis to the new Figure 7—figure supplement 1.

*7) The PWMs in the supplement table is missing. This information should be made publically available*.

We apologize for including the PWM data in an unusual format. We have now included the individual PWMs as a [Supplementary-material SD6-data], and as an Excel spreadsheet with annotation ([Supplementary-material SD5-data]).